# Evaluation of *Bacillus velezensis* F9 for Cucumber Growth Promotion and Suppression of *Fusarium wilt* Disease

**DOI:** 10.3390/microorganisms12091882

**Published:** 2024-09-12

**Authors:** Yongquan Ta, Shaowei Fu, Hui Liu, Caiyun Zhang, Mengru He, Hang Yu, Yihua Ren, Yunfei Han, Wenqiong Hu, Zhiqiang Yan, Yonghong Wang

**Affiliations:** 1Key Laboratory of Plant Protection Resources and Pest Management of Ministry of Education, Key Laboratory of Integrated Pest Management on the Loess Plateau of Ministry of Agriculture and Rural Affairs, College of Plant Protection, Northwest A&F University, Yangling 712100, China; yongquanta@163.com (Y.T.); 2019010331@nwafu.edu.cn (S.F.); 13310648798@163.com (H.L.); zhangcaiyun910@163.com (C.Z.); 17391708622@163.com (M.H.); yuhang9604@163.com (H.Y.); renyihua@nwafu.edu.cn (Y.R.); hanyunfei@nwafu.edu.cn (Y.H.); huwenqiong@163.com (W.H.); yan0517@nwafu.edu.cn (Z.Y.); 2Provincial Center for Bio-Pesticide Engineering, Northwest A&F University, Yangling 712100, China

**Keywords:** *Bacillus velezensis* F9, antifungal activity, plant growth promotion, disease suppression, biological control, cucumber wilt

## Abstract

Cucumber wilt, caused by *Fusarium oxysporum f*. sp. *cucumerinum* (FOC), is a soilborne disease that poses a significant threat to cucumber production, resulting in substantial yield losses. This study aimed to evaluate the biocontrol and growth-promoting effects of *Bacillus velezensis*, a highly active bacterial strain. In vitro assays revealed that *B. velezensis* F9 exhibited broad-spectrum antifungal activity against eight plant pathogenic fungi, with inhibition ratio ranging from 62.66% to 88.18%. Additionally, the strain displayed the ability to produce IAA (5.97 ± 1.75 µg/mL), fix nitrogen, produce siderophores, and form biofilms. In vitro growth promotion assays demonstrated that different concentrations of *B. velezensis* F9 significantly promoted cucumber seedling growth. Furthermore, two pot experiments revealed that the strain exhibited biocontrol efficacy against cucumber wilt, with disease control rates ranging from 42.86% to 67.78%. Notably, the strain significantly increased the plant height, fresh weight, and dry weight, with increases ranging from 20.67% to 60.04%, 40.27% to 75.51%, and 22.07% to 52.54%, respectively. Two field trials confirmed the efficacy of *B. velezensis* F9 in controlling cucumber wilt, with disease control rates of 44.95% and 33.99%, respectively. The strain effectively alleviated the dwarfing and wilting symptoms caused by the pathogen. Compared with the FOC treatment, the F9 + FOC treatment significantly increased the plant height, fresh weight, and dry weight, with increases of 43.85% and 56.28%, 49.49% and 23.70%, and 36.25% and 73.63%, respectively. Enzyme activity assays indicated that inoculation significantly increased SOD activity in cucumber leaves and neutral phosphatase, sucrase, and urease activity in rhizosphere soil. Correlation analysis revealed a negative correlation between the disease index and plant height, fresh weight, dry weight, and peroxidase activity, with correlation coefficients of −0.53, −0.60, −0.38, and −0.45, respectively. These findings suggest that plant height, fresh weight, and dry weight are significantly negatively correlated with the cucumber disease index, highlighting their importance as indicators for evaluating the biocontrol efficacy of *B. velezensis* F9. In conclusion, *B. velezensis* F9 is a highly effective plant growth-promoting rhizobacterium with excellent biocontrol potential, showcasing promising applications in agricultural production.

## 1. Introduction

Cucumber (*Cucumis sativus* L.) is a common vegetable and commercial crop worldwide [1]. In 2020, cucumber cultivation reached 2.25 million hectares worldwide, with a total yield of 90.35 million tons. China leads in terms of cucumber cultivation area and output, with 1.27 million hectares and 73.36 million tons, respectively [2]. However, in recent years, cucumber wilt disease has caused a 10–20% yield loss in China, with total crop collapse in severe cases as the number of years of cultivation has increased [3]. Cucumber wilt disease is a typical soilborne disease caused by *Fusarium oxysporum f.* sp. *cucumerinum* (FOC), which infects cucumbers throughout their growth period by invading from roots and stems and parasitizing the vascular bundles, thereby hindering the water and nutrient absorption of the plant, resulting in wilting and withering of the cucumber [4].

Currently, cucumber varieties resistant to cucumber wilt disease are still limited. Traditional management measures, including chemical control and soil fumigation, often lead to environmental pollution, pathogen resistance, and food safety issues [5]. Additionally, agricultural measures such as grafting and crop rotation can also be used to control cucumber wilt disease, but they have limitations such as complexity, labor intensity, and high costs. Biological control has become the most effective measure for managing soilborne diseases, helping to protect farmland ecosystems and ensuring the safety of humans and animals while effectively reducing the emergence of resistant pathogens [6].

*Bacillus* species are extensively distributed in nature and within plants, displaying effective colonization of the plant rhizosphere. Through spore production, these organisms demonstrate resilience against adverse environmental conditions, including high temperature, ultraviolet radiation, and extreme pH levels. *Bacillus* produces abundant green, nontoxic, and environmentally friendly secondary metabolites that inhibit plant diseases. It is considered an ideal biocontrol microorganism for plant disease management [2]. Currently, many *Bacillus* strains, including *Bacillus subtilis* [7,8,9], *Bacillus amyloliquefaciens* [10,11], *Bacillus velezensis* [12,13,14], *Bacillus atrophaeus* [15], and *Bacillus siamensisi* [16], are used to control cucumber wilt. However, microorganisms from different sources vary in morphology and physiological and biochemical characteristics. These strains often have single functional traits and lack multifunctional characteristics for further development and utilization. Thus, isolating and screening highly active strains from different habitats can improve the efficacy of *Bacillus* biocontrol against cucumber wilt.

Our research group previously isolated and identified rhizosphere microorganisms from wheat and evaluated their antibacterial and growth-promoting effects. We found that strain F9 exhibited good antibacterial and growth-promoting characteristics. In this study, we first verified the antibacterial and growth-promoting activity of strain F9 and further identified the strain via molecular biology techniques. Additionally, we conducted a comprehensive evaluation of the effects of different inoculation methods (preventive, therapeutic, and simultaneous) and different active forms (bacterial suspension and deactivated fermentation broth) of strain F9 on cucumber wilt disease and the growth-promoting activity of cucumber plants across two experimental batches. We further validated the effectiveness of strain F9 in controlling cucumber wilt in a small plot trial. Finally, by measuring peroxidase (POD) and superoxide dismutase (SOD) activities in the leaves of cucumber plants in potted and small plot trials, we evaluated the impact of strain F9 on cucumber defense enzyme activity. We also measured soil alkaline phosphatase, urease, and sucrase activities in both potted and small plot trials to assess the influence of strain F9 on soil enzyme activity. The assessment of these indicators allows for a comprehensive evaluation of the disease prevention and growth promotion effects of strain F9 and lays the foundation for research into its mechanisms of action. The overall experimental procedure is shown in Figure 1.

## 2. Materials and Methods

### 2.1. Fungal Pathogens, Biocontrol Strains and Plants

Fungal Pathogens: *Rhizoctonia solani*; *Sclerotinia sclerotiorum*; *Botrytis cinerea*; *Gaeu-mannomyces graminsis (sacc.) Arx & Olivier Var tritici J. Walker*; *Alteraria alternata*; *Botryospuaeria berengeriana*; *Fusarium oxysporum f.* sp. *Niveum*; and *Fusarium oxysporum.* sp. *cucumerinum* were provided by the Provincial Center for Bio-Pesticide Engineering, Northwest A&F University. The strains were cryopreserved in potato dextrose broth (PDB) medium containing 20% glycerol at −80 °C. All of them were routinely cultured on potato dextrose agar (PDA) media at 28 °C.

Biocontrol Strain: The strain F9 isolated from the wheat rhizosphere was provided by the Provincial Center for Bio-Pesticide Engineering, Northwest A&F University, which was cryopreserved in LB medium containing 20% glycerol at −80 °C.

Plant: The variety of cucumber used was Jingyan No. 4, which was purchased from the seed market of Yangling.

### 2.2. Plant Growth Promotion (PGP) Traits Evaluation

#### 2.2.1. Indole-3-Acetic Acid Production

The production of indole-3-acetic acid (IAA) was performed following the protocol of Borah et al. [17] with some modifications. Isolates were inoculated into LB medium (10 g of tryptone, 5 g of yeast extract, 5 g of NaCl, per liter, 1000 mL of distilled water, pH = 7.0–7.2) containing 100 µg/mL of L-tryptophan and incubated at 28 °C with continuous agitation at 200 rpm for 48 h. Approximately 2 mL aliquots of the cultures were centrifuged for 5 min at 8000 rpm, and 100 µL of the supernatant was mixed with 200 µL of Salkowski reagent (1 mL of 0.5 M FeCl_3_ in 50 mL of 35% HClO_4_) in a 96-well microplate, which was incubated in the dark for 30 min at room temperature. Uninoculated sterile media was used as a blank control. The absorbance of the pink-red color was determined at 530 nm via a multimode reader (SpectraMax^®^ Plus384, La Jolla, CA, USA), and the concentration of IAA was calculated via a standard curve of commercial IAA (0, 0.625, 1.25, 2.5, 5, 10, and 50 µg/mL).

#### 2.2.2. Siderophore Production

A chrome azazrol assay was carried out to determine the siderophore activity of the bacterial isolates [18]. For primary screening, 48-h-old bacterial cultures grown in LB media were spotted on CAS (Chrome Azurol S) agar plates and incubated at 28 °C for 7 days. The formation of a yellow-orange halo around the isolate colonies was considered an indication of siderophore production.

#### 2.2.3. Potassium Solubilization

The potassium solubilization ability of the strains was evaluated according to the methods of Sarikhani et al. [19]. Using the spot inoculation method, activated strains were inoculated onto Aleksandrov medium (5 g of glucose, 0.5 g of MgSO_4_·7H_2_O, 0.0005 g of FeCl_3_, 0.1 g of CaCl2, 0.2 g of Al_2_K_2_Na_2_O_15_Si_5_, 0.2 g of Ca_3_(PO_4_)_2_, 1000 mL of distilled water, pH = 7.0–7.2) and incubated at a constant temperature of 28 °C for 5 days. The ability of the strains to solubilize potassium was evaluated on the basis of the presence and size of clear zones around the colonies.

#### 2.2.4. Biological Nitrogen Fixation

The strain F9 samples were inoculated onto ACCC55 nitrogen-free medium (10 g of Sucrose, 0.12 g of NaCl, 0.5 g of K_2_HPO_4_·3H_2_O, 1 g of CaCO_3_, 0.2 g of MgSO_4_·7H_2_O, 20 g of agar, 1000 mL of distilled water, pH = 7.0–7.2) plates to observe their growth ability [20].

### 2.3. Biofilm Formation

The biofilm-forming ability of the strains was evaluated via the methods of Shemesh and Chai [21]. Strain F9 was cultured in LB media to a concentration of 10^9^ cfu/mL. In a 24-well plate, 2 mL of LB medium, LBG medium (LB plus 1% [*v*/*v*] glycerol), LBM medium (LB plus 0.1 mM of MnSO_4_), or LBGM medium (LB plus 1% [*v*/*v*] glycerol and 0.1 mM of MnSO_4_) was added, and 2 μL of bacterial suspension was placed onto the surface of LB, LBG, LBM, or LBGM media. After 3 days, the formation of a biofilm on the liquid surface was observed.

### 2.4. Molecular Identification of Functional Bacteria

The genomic DNA of all positive isolates was extracted with a Rapid Bacterial Genomic DNA isolation kit (Sangon Biotech, Shanghai, China). The 16S rRNA gene was amplified via the universal primers 27F (5′-GAGTTTGATCACTGGCTCAG-3′) and 1492R (5′-TACGGCTACCTTGTTAC GACTT-3′) and sequenced by Shannxi Aoke Biotechnology Co., Ltd. (Shannxi, China). The 16S rRNA gene sequences were compared with the National Center for Biotechnology Information (NCBI) GenBank database via BLAST tools.

### 2.5. Effects of Bacterial Seed Treatment on the Growth of Cucumber Plants

First, the strains were inoculated into 250 mL conical flasks containing 100 mL of LB liquid medium, with each strain being inoculated twice. The flasks were incubated at 28 °C and 180 r/min for 72 h. Then, 8 mL of the culture broth was centrifuged at 8000 r/min and 4 °C, the supernatant was discarded, and the cells were washed three times with sterile PBS buffer. Finally, the cells were resuspended in PBS (phosphate buffer saline) to an OD_600_ of 1.0 and stored at 4 °C for later use.

Next, the cucumber seeds were sterilized with 75% anhydrous ethanol for 10 min, rinsed three times with sterile water, sterilized with 2% sodium hypochlorite solution for 10 min, rinsed three times with sterile water, and stored for later use.

Twenty sterilized seeds were placed in a Petri dish (9 cm) lined with sterile filter paper, and the prepared bacterial suspension was added. All the plants were placed in the dark for 3 days, after which the germination ratio was measured. The seedlings were then cultured under light (18/6 L/D) for 7 days, after which the cucumber plant height and root length were measured. All the treatments were repeated twice.

### 2.6. The Biocontrol Efficacy of Strain F9 against Cucumber Wilt Disease in Pot and Plot Experiments

The pot experiment was conducted in a solar greenhouse at the North Campus of Northwest A&F University (34°17′ N, 108°04′ E) in Yangling, Shaanxi Province. The potting soil was a mixture of field soil and substrate at a ratio of 1:1, where the field soil was collected from the wheat experimental field at the North Campus of Northwest A&F University, air-dried, and sieved through a 2 mm mesh.

The strain was cultured in LB medium at 28 °C and 180 rpm for 72 h., resulting in fermentation broth with an OD_600_ of 1.0. The fermentation broth was centrifuged at 8000 rpm for 5 min, and the resulting supernatant was used for the experiments (F9_F). The bacterial pellet was resuspended in an equal volume of sterile water to obtain a bacterial suspension of 1.0 × 10^8^ CFU/mL, which was used for the experiments (F9_C). This experiment used three inoculation methods: simultaneous inoculation of the pathogen and biocontrol agent (B), inoculation of the pathogen one week before the biocontrol agent (C), and inoculation of the biocontrol agent one week before the pathogen (P).

Cucumber seeds were sown in a seedling tray for seedling production. After one week, the seedlings were transplanted into pots or plots, with one seedling per pot. After the cucumbers grew to the 2-leaf-1-heart stage, their biocontrol efficacy was evaluated via the wounded root dipping and root irrigation methods. The experimental treatments included the following: (1) inoculation with only 25 mL of cucumber wilt pathogen and an equal amount of sterile water (FOC); (2) co-inoculation with 25 mL of cucumber wilt pathogen and an equal amount of the isolated bacterial strain suspension (10^8^ CFU/mL); (3) inoculation with only 25 mL of the F9 bacterial suspension (10^8^ CFU/mL) and an equal amount of sterile water; (4) co-inoculation with 25 mL of cucumber wilt pathogen and an equal amount of supernatant; (5) inoculation with 25 mL of the F9 bacterial supernatant and an equal amount of sterile water; and (6) inoculation with 25 mL of sterile water (CK). Each treatment was repeated four times. According to Xu et al. [9], one month after inoculation, the disease index, leaf chlorophyll content, plant height, stem diameter, and fresh weight and dry weight of aboveground biomass were recorded and the biocontrol efficacy was calculated. In brief, disease severity was assessed using a 0–4 disease scale; 0 = leaf asymptomatic; 1 = leaf wilting below 1/4 of cucumber seedling; 2 = leaf wilting in 1/4 to 1/2 of cucumber seedling; 3 = leaf wilting above 1/2 of cucumber seedling; 4 = the whole plant was wilted and died. The disease index was calculated using DI = [[(0 × N0) + (1 × N1) + (2 × N2) + (3 × N3) + (4 × N4)]/T × 4] × 100, where N is the number of cucumber seedlings for each disease score, and T is the total number of cucumber seedlings. Control efficacy = (DI of control − DI of treatment)/DI of control × 100%.

In June 2020, an experiment was conducted to evaluate the biocontrol efficacy of rhizosphere microbes against cucumber wilt disease and their plant growth-promoting effects on cucumbers. In this experiment, the treatments were 1, 2, 3, 4, 5, and 6. In June 2021, further verification of the biocontrol efficacy of strain F9 against cucumber wilt disease and its growth-promoting effects on cucumbers, which included treatments 1, 2, 3, and 6, was carried out. Additionally, in the same greenhouse, plot experiments were conducted in June and July 2021 to evaluate the biocontrol efficacy of strain F9 against cucumber wilt disease. The treatments of the plot experiments included 1, 2, 3, and 6.

### 2.7. Measurement of Defense Enzyme Activities in Cucumber Leaves in Plots

The total SOD and POD activity was determined according to the method described by Meloni et al. [22].

The plant leaf SOD activity was determined via the inhibition of nitro blue tetrazolium reduction. Briefly, fresh leaf samples (usually 0.5 g) were homogenized in phosphate buffer (pH 7.8) under ice-bath conditions. After centrifugation (12,000 rpm, 20 min), the supernatant was used as the enzyme extract. The reaction system included phosphate buffer, methionine, nitro blue tetrazolium, riboflavin, and the enzyme extract. The absorbance at 560 nm was measured under light conditions. The enzyme activity was calculated, and one unit (U) was defined as the amount of enzyme required to inhibit 50% of the nitro blue tetrazolium reduction per minute. The formula is as follows:$$SOD\;activity\;(U\cdot g^{-1}FW\cdot h^{-1})=\frac{\left(A_{O}-A_{S}\right)\times V_{t}\times 60}{A_{O}\times 0.5\times FW\times V_{s}\times t}$$
where: A_o_—Absorbance of the control tube at 560 nm under light conditions; A_s_—Absorbance of the sample tube at 560 nm; V_t_—Total volume of sample extract (mL); V_s_—Volume of crude enzyme solution used for measurement (mL); FW—Fresh weight of the sample (g); t—Duration of color reaction under light (min).

The POD activity of the plant leaves was determined via hydrogen peroxide-dependent oxidation of o-phenylenediamine. Fresh leaf samples (usually 0.5 g) were homogenized in phosphate buffer (pH 6.0) under ice-bath conditions. After centrifugation (12,000 rpm, 20 min), the supernatant was used as the enzyme extract. The reaction system included phosphate buffer, hydrogen peroxide, o-phenylenediamine, and the enzyme extract. The change in absorbance at 470 nm was measured. The enzyme activity was calculated, and one unit (U) was defined as the amount of enzyme required to oxidize 1 micromole of substrate per minute. The formula is as follows:$$POD\;activity\;(U\cdot g^{-1}\cdot {min}^{-1})=\frac{△A_{470}\times V_{t}}{FW\times V_{s}\times 0.01\times t}$$
where: ΔA_470_—Change in absorbance over reaction time; FW—Fresh weight of leaves (g); t—Reaction time (min); V_t_—Total volume of enzyme extract (mL); V_S_—Volume of enzyme solution used for measurement (mL).

### 2.8. Measurement of Soil Enzyme Activities in Plots

The neutral phosphatase (S-NP) activity in the soil was determined via a soil-neutral phosphatase activity detection kit. The formula is as follows: Phosphatase activity = (A_sample − A_soil-free − A_matrix-free) × V × n/m, where A_sample is phenol concentration (mg) in the sample, determined from the standard curve using the sample absorbance value; A_soil-free is phenol concentration (mg) in the soil-free control, determined from the standard curve using the soil-free control absorbance value; A_matrix-free is phenol concentration (mg) in the matrix-free control, determined from the standard curve using the matrix-free control absorbance value; V is volume of color reagent (mL); n is dilution factor, calculated as the volume of extractant solution/volume of filtered solution; m is Dry weight of soil (g).

An appropriate amount of air-dried soil sample (usually 1 g) was added to a sucrose solution and buffer (pH 5.5). The mixture was incubated at 37 °C for 24 h After the reaction, chloroform was added to terminate the reaction, and the mixture was filtered. The reducing sugar content was measured via colorimetry at 540 nm. The enzyme activity was calculated and defined as the amount of reducing sugars (mg) produced per gram of soil per h. The formula is as follows: Urease activity = (A_sample − A_soil-free − A_matrix-free) × V × n/m, where A_sample is NH_3_-N concentration (mg) in the sample, determined from the standard curve using the sample absorbance value; A_soil-free is NH_3_-N concentration (mg) in the soil-free control, determined from the standard curve using the soil-free control absorbance value; A_matrix-free is NH_3_-N concentration (mg) in the matrix-free control, determined from the standard curve using the matrix-free control absorbance value; V is volume of color reagent (mL); n is dilution factor, calculated as the volume of extractant solution/volume of filtered solution; m is dry weight of soil (g).

An appropriate amount of air-dried soil sample (usually 1 g) was added to a urea solution and buffer (pH 7.0). The mixture was incubated at 37 °C for 2 h. After the reaction, chloroform was added to terminate the reaction, and the mixture was filtered. The ammonia content was measured via colorimetry at 630 nm. The enzyme activity was calculated and defined as the amount of ammonia (mg) produced per gram of soil per hour. The formula is as follows: Sucrase activity = (A_sample − A_soil-free − A_matrix-free) × n/m, where A_sample is glucose concentration (mg) in the sample, determined from the standard curve; A_soil-free is glucose concentration (mg) in the soil-free control, determined from the standard curve; A_matrix-free is glucose concentration (mg) in the matrix-free control, determined from the standard curve; n is dilution factor; m is dry weight of soil (g).

### 2.9. Statistical Analyses

All the collected data were analyzed via ANOVA in R 4.3.1 software. The data obtained from the replicates of each experiment are presented in the graphs as the means ± standard errors (SEs). The mean values were compared using LSD multiple range test with a significance level set at *p* < 0.05. “corrplot” package in R was used to calculate the Pearson correlations among cucumber plant phenotypic traits and draw the correlation heatmap.

## 3. Results

### 3.1. Evaluation of the Antagonistic and Plant Growth-Promoting Characteristics of Strain F9

Further verification of the antagonistic activity, plant growth-promoting functions, and biofilm-forming ability of the F9 strain was carried out. The results revealed that the F9 strain had significant antagonistic activity against the pathogens of rice sheath blight (66.39%), oilseed rape Sclerotinia stem rot (88.18%), tomato gray mold (85.24%), wheat take-all disease (64.07%), tobacco black shank (68.01%), apple ring rot (66.55%), watermelon Fusarium wilt (62.66%), and cucumber Fusarium wilt (64.07%) (Figure 2a–p, Table 1). In addition, the F9 strain also produced IAA (5.97 ± 1.75 µg/mL), fixed nitrogen, produced siderophores, and formed biofilms (Figure 2q–x, Table 2).

On the basis of the 16S rDNA sequence, a phylogenetic tree of the F9 strain was constructed via the neighbor-joining method in MEGA 7.0 software. The results revealed that the F9 strain was most similar to the *Bacillus velezensis* strain TPS3N (MK130897.1); thus, the F9 strain was identified as *Bacillus velezensis*. The gene sequence has been deposited in the GenBank database with the accession number MW314755 (Figure 3). This strain was deposited in the China General Microbiological Culture Collection Center (CGMCC) on 7 December 2020, with the strain number CGMCC No. 21318.

### 3.2. Growth Promotion Efficiency of Strain F9 on Cucumber In Vitro

To investigate the growth-promoting effect of strain F9 on cucumber seedlings, a study was conducted to assess the impact of four different concentrations (10^8^, 10^7^, 10^6^, and 10^5^ cfu/mL) of the strain on the seed germination ratio, shoot length, root length, and fresh weight (Figure 4a). The results indicated that strain F9 significantly promoted the growth of cucumber seeds at concentrations of 10^8^ cfu/mL (96.67%), 10^7^ cfu/mL (100%), 10^6^ cfu/mL (100%), and 10^5^ cfu/mL (93.33%). However, no significant difference (*p* > 0.05) was detected compared with the control seed germination ratio (93.33%). Compared with the control, strain F9 significantly promoted shoot length at a concentration of 10^7^ cfu/mL (45.08 ± 2.5 cm) (*p* < 0.05), resulting in a 21.05% increase in shoot length. While concentrations of 10^8^ cfu/mL (41.98 ± 2.67 cm), 10^6^ cfu/mL (38.96 ± 2.54 cm), and 10^5^ cfu/mL (43.04 ± 2.69 cm) also had positive effects on shoot length, these differences were not statistically significant (*p* > 0.05). With respect to root length, all concentrations except 10^8^ cfu/mL (80.28 ± 5.07 cm) significantly promoted cucumber seedling root length (*p* < 0.05). The respective root length growth rates at concentrations of 10^7^ cfu/mL (96.53 ± 4.72 cm), 10^6^ cfu/mL (98.80 ± 4.80 cm), and 10^5^ cfu/mL (89.38 ± 5.07 cm) were 55.69%, 59.35%, and 44.16%, respectively. Furthermore, strain F9 significantly increased cucumber seedling fresh weight at all concentrations tested (*p* < 0.05), with fresh weight growth rates ranging from 49.47% to 61.76%. In conclusion, these findings demonstrate that strain F9 has a significant growth-promoting effect on cucumber seedlings, particularly in terms of shoot and root length and fresh weight.

### 3.3. Biocontrol Efficiency of Strain F9 on Cucumber Fusarium Wilt in Pots

In June 2020, the efficacy of the *Bacillus velezensis* strain F9 in controlling cucumber wilt was evaluated. 

The results revealed that simultaneous inoculation of strain F9 and the cucumber wilt pathogen had the strongest disease control effect, with F9_C_B (58.33%) and F9_F_B (50.00%) being the most effective. Inoculation with the pathogen first followed by the biocontrol agent had a moderate effect, with F9_C_C (33.33%) and F9_F_C (45.00%). The biocontrol agent was inoculated first, and then the pathogen had the weakest effect, with F9_C_P (5.66%) and F9_F_C (6.00%). There was no obvious difference in the disease control effect between the bacterial cells and the cell-free fermentation broth (Figure 5a).

Compared with inoculation with the cucumber wilt pathogen alone (FOC), all the inoculation methods with strain F9 significantly increased the chlorophyll content of the cucumber plants (*p* < 0.05), with increases ranging from 18.20% to 74.02%, except for the F9_C_C treatment (Figure 5b).

Furthermore, compared with the FOC treatment, simultaneous inoculation with F9 and the pathogen (F9_C_B and F9_F_B) significantly increased the height (56.79% and 60.04% increase, respectively), fresh weight (73.56% and 75.51% increase, respectively), and dry weight (45.20% and 52.54% increase, respectively) of the cucumber plants. The stem diameter was also reduced (Figure 5c–f).

In June 2021, a pot trial was conducted to determine the effects of the *Bacillus velezensis* strain F9 on the control of cucumber wilt disease and its growth-promoting activity on cucumber plants.

With respect to disease control, the F9 strain had a disease control efficacy of 42.86% against cucumber wilt. Compared with the control treatment, the F9_FOC treatment resulted in significantly greater plant height (112.50 cm), fresh weight (25.50 g), and dry weight (3.92 g) of the cucumber plants (89.25 cm, 15.23 g, and 3.055 g, respectively), with increases of 20.67%, 40.27%, and 22.07%, respectively. These findings indicate that strain F9 can significantly alleviate the stunting of cucumber plants caused by the wilt pathogen (Figure 6a–c).

With respect to growth promotion, the F9 strain did not have a significant growth-promoting effect on the cucumber plants, as there were no significant differences in plant height, stem diameter, fresh weight, or dry weight compared with those of the control (Figure 6a,c–f).

### 3.4. Biocontrol Efficiency of Strain F9 on Cucumber Fusarium Wilt in Plots

In June 2021, the field efficacy of the *Bacillus velezensis* strain F9 in controlling cucumber wilt was evaluated. The results revealed that the field control efficacy of strain F9 against cucumber wilt was 44.95%. The plant height, stem diameter, fresh weight, and dry weight of the plants in the F9 + FOC treatment were comparable to those in the blank control (CK), and the plant height, fresh weight, and dry weight were significantly greater than those in the FOC treatment, with increases of 43.85%, 49.49%, and 36.25%, respectively (Figure 7a,b; Table 3).

In July 2021, the efficacy of F9 in controlling cucumber wilt was tested in a different plot within the same greenhouse. The results revealed that the field control efficacy of F9 against cucumber wilt was 33.99%. The plant height, stem diameter, fresh weight, and dry weight in the F9 + FOC treatment were comparable to those in the blank control (CK), and they were all significantly greater than those in the FOC treatment, with increases of 56.28%, 23.13%, 23.70%, and 73.63%, respectively (Figure 7a,b; Table 4).

Furthermore, the soil enzyme activities in the cucumber rhizosphere from the June 2021 greenhouse plot trial were measured. The results revealed that the F9_FOC treatment had no significant effect on the activities of neutral phosphatase, sucrase, or urease in the cucumber soil, but the activity of alkaline phosphatase in the cucumber rhizosphere was greater in the F9_FOC treatment than in the blank control and the pathogen control (Table 5).

The defense enzyme activities in the leaves of the cucumber plants from the two greenhouse plot trials were further measured. The results revealed that inoculation with the F9 strain increased superoxide dismutase activity in cucumber leaves, but the difference was not significant. Additionally, inoculation with the F9 strain reduced peroxidase activity in cucumber leaves, and peroxidase activity was significantly lower than that in the control (*p* < 0.05) in the July 2021 plot trial (Table 6).

### 3.5. Correlation Analysis

Correlation analysis was conducted on the indicators from the greenhouse plot trials, including the cucumber plant disease index; plant height; stem diameter; fresh weight; dry weight; neutral phosphatase, sucrase and urease activities in the cucumber rhizosphere; and peroxidase and superoxide dismutase activities in the cucumber leaves.

As shown in Figure 8, the cucumber plant disease index was positively correlated with soil sucrase activity but negatively correlated with other cucumber plant phenotypic indicators. Specifically, the disease index had relatively strong negative correlations with plant height, fresh weight, dry weight, and peroxidase activity, with correlation coefficients of −0.53, −0.60, −0.38, and −0.45, respectively.

Furthermore, most of the cucumber plant phenotypic indicators were positively correlated with each other. Plant height, fresh weight, and dry weight were significantly positively correlated (*p* < 0.01), with correlation coefficients of 0.91, 0.87, and 0.93, respectively. SOD activity was also significantly positively correlated with plant height and fresh weight (*p* < 0.05), with correlation coefficients of 0.79 and 0.70, respectively. Urease was positively correlated with plant height and neutral phosphatase activity, with correlation coefficients of 0.70 (*p* < 0.05) and 0.67 (*p* < 0.01), respectively.

## 4. Discussion

In recent years, microbiome studies based on high-throughput sequencing technology have shown that the plant rhizosphere microbiome can act as the “second genome” of plants and play an important role in plant health. This concept has led to a new understanding of the crucial role of soil microorganisms (including plant rhizosphere microbes) in determining plant phenotypes [23]. Plant growth-promoting rhizobacteria (PGPRs) are a group of microorganisms that colonize plant roots and benefit plant growth and immunity, and they have received much attention because of their protective effects on crops and ecosystems. Many microorganisms in the rhizosphere are called plant growth-promoting rhizobacteria, including fast-growing and motile *Pseudomonas* with diverse metabolites, as well as *Streptomyces*, which can produce large amounts of antibiotics. However, the main obstacle to the application of *Streptomyces* and *Pseudomonas* in agricultural production is the short shelf-life of their liquid fermentation products and their short survival time in soil [24]. *Bacillus* and *Paenibacillus* can compensate for these shortcomings of *Pseudomonas* and *Streptomyces* and are promising candidates among plant growth-promoting rhizobacteria [25]. In addition to their longer survival cycle, *Bacillus* species also have the ability to produce IAA, ACC, and siderophores; solubilize organic and inorganic phosphorus; solubilize potassium; fix nitrogen; produce various enzymes; produce large amounts of peptide antibiotics; and have been widely used in the development of microbial fertilizers to promote plant growth, enhance plant stress resistance, and help plants resist disease invasion [26].

Owing to the limitations of early bacterial identification techniques, “heterotypic synonyms” are common in the *Bacillus* genus, leading to confusion between closely related species. With the gradual maturation of microbial taxonomy and identification techniques, some *Bacillus* species have been renamed, including the taxonomic changes and determinations of *Bacillus velezensis* [27]. According to the results of DNA–DNA hybridization, *Bacillus velezensis* is a late heterotypic synonym of *Bacillus amyloliquefaciens* [28]. In addition, microorganisms living in different environments undergo continuous genetic mutations during the process of environmental adaptation and can also undergo horizontal gene transfer (HGT) by integrating DNA fragments or plasmids from other organisms in the environment, forming specific plasmids and genomic islands [29]. A typical example is *Bacillus velezensis* SQR9, which acquires a genomic island through horizontal gene transfer, encoding the synthesis of a new antibiotic, bacilysocin, which can inhibit other *Bacillus* in the environment, thereby providing protection for *Bacillus velezensis* SQR9 [30]. Strains of the same type that exist in different environments or different subtypes identified as the same species often have their own functional characteristics. Extensive research efforts have been devoted to the isolation, identification, and functional evaluation of biocontrol Bacillus strains from diverse environments, providing a strong foundation for the biological control of plant diseases and sustainable agricultural practices. This study focused on *Bacillus velezensis* F9, a plant growth-promoting rhizobacterium isolated from wheat rhizosphere soil. Understanding its biocontrol efficacy and mechanism is crucial for developing effective strategies for plant disease management.

*Bacillus velezensis* was first discovered and named by the Spanish scholars Ruiz-García et al. in 2005; it was previously considered a subspecies of *Bacillus subtilis* and is closely related to *Bacillus amyloliquefaciens* [31]. *Bacillus velezensis* has been widely used in winemaking and food fermentation [32,33], animal feed additives and silage [34], fish disease control [35], soil remediation [36], plant growth promotion, stress resistance, and disease control [37]. Among them, research on plant growth promotion and disease control has been more extensive. For example, *Bacillus velezensis* strain NKG-2 isolated from Qilian Mountains in Qinghai province reduced the disease index of tomato *Fusarium wilt* from 65% to 25% [38]; *Bacillus velezensis* strain C2 isolated from tomato crown tissue achieved 70.43 ± 7.08% potting control efficacy against tomato Verticillium wilt [39]; *Bacillus velezensis* strain AR1 reduced the disease index of sesame *Alternaria* leaf spot from 83.3% to 50% in potting [40]; *Bacillus velezensis* strain BV01 reduced the disease index of wheat root rot from 76.4% to 40.8% in potting [41]; the solid fermentation product of *Bacillus velezensis* strain NH-1 achieved 85.96 ± 3.04% control efficacy against cucumber *Fusarium wilt* under potting conditions, and the sodium alginate microcapsule formulation increased the control efficacy by 14.04% to 100%; the bioorganic fertilizer prepared by *Bacillus velezensis* strain SQR9 reduced the disease index of cucumber *Fusarium wilt* from 73% to 28%, with a control efficacy of 61.64%, and the bioorganic fertilizer prepared jointly with *Paenibacillus polymyxa* strain SQR-21 and *Trichoderma harzianum* strain SQR-T037 achieved 83% field control efficacy against cucumber *Fusarium wilt* [14,42]. This study revealed that the *Bacillus velezensis* strain F9 achieved 42.86–67.78% potting control efficacy and 44.95% and 33.99% field control efficacy against cucumber *Fusarium wilt* in different batches, which is equivalent to the control efficacy of *Bacillus velezensis* alone but significantly weaker than the control efficacy of the *Bacillus velezensis* microcapsule formulation and multistrain bioorganic fertilizer.

The direct inhibition of plant pathogens is an important mechanism by which plant growth-promoting rhizobacteria exert their control effects. *Bacillus velezensis* strain NKG-2 has inhibition activities of 60%, 56.47%, 50.0%, 50.0%, 50.67%, and 52.38% against *Fusarium oxysporum*, *Fusarium graminearum*, *Botrytis cinerea*, *Alternaria alternata*, *Fulvia fulva*, and *Ustilaginoidea* virens, respectively [38]. *Bacillus velezensis* strain BV01 has high inhibition activities, ranging from 57% to 83%, against various pathogens, such as *Bipolaris sorokiniana*, *Botrytis cinerea*, *Colletotrichum capsici*, *Fusarium graminearum*, *Fusarium oxysporum*, *Neocosmospora rubicola*, *Rhizoctonia solani*, and *Verticillium dahlia* [41]. This study revealed that the *Bacillus velezensis* strain F9 has a broad-spectrum inhibitory effect on various plant pathogens, with inhibitory activities ranging from 62.66% to 88.18%, which is consistent with the above studies. The production of many inhibitory secondary metabolites, including lipopeptides such as surfactin, fengycin, and bacillomycin; polyketides such as macrolides, polyenes, difficidin, and hydroxydifficidin; and peptides such as plantazolicin, amylocyclicin, and bacily, is one of the reasons why *Bacillus velezensis* can inhibit various plant pathogenic fungi [43]. The production of many hydrolytic enzymes that can degrade fungal cell walls, such as β-1,3-glucanase, chitinase, and cellulase, is another reason for the inhibition of plant pathogens by *Bacillus velezensis* [38].

The growth-promoting effect of the *Bacillus velezensis* strain F9 on cucumber plants may be related to its ability to produce IAA, siderophores, and nitrogen fixation, as well as its ability to form biofilms. IAA, a major plant growth hormone, is closely related to the physiological and biochemical processes of plant growth and development and is used to regulate plant cell division, elongation, and differentiation, as well as root and stem growth [44]. *Bacillus velezensis* strain C2 produces 38.85 ± 1.6 μg/mL IAA in the presence of 1 mg/mL of tryptophan, whereas *Bacillus velezensis* strain F9 in this study produces 5.97 ± 1.75 µg/mL of IAA in the presence of 100 μg/mL of tryptophan. The content of siderophores is positively correlated with plant growth, and the siderophores secreted by plant growth-promoting rhizobacteria can capture more iron under low-iron conditions to support plant growth. In addition, siderophores produced by microorganisms also contribute to the colonization of plant endophytes [45]. The siderophores produced by microorganisms can also reduce the absorption of iron by pathogens in the plant rhizosphere environment to achieve disease control [46]. A biofilm is a community of microbial cells encased in a self-produced extracellular matrix that is primarily composed of exopolysaccharides, proteins, and DNA. The biofilm matrix provides structural support and protection for microbial cells, enabling them to adhere to surfaces and bind to each other. Biofilms offer numerous advantages to microorganisms, including enhanced resistance to antimicrobial agents and host immune responses, acquisition of new genetic traits through horizontal gene transfer, retention of nutrients and moisture within the matrix, and protection from predators and environmental stresses [47]. The biocontrol efficacy of biocontrol agents is closely related to their ability to colonize plant surfaces. Biofilm formation plays a crucial role in the initial stages of biocontrol. In the control of soilborne diseases, bacterial biofilms facilitate the attachment of biocontrol agents to roots, providing nutrients for their growth. *Bacillus velezensis* HYEB5-6 exhibits strong biofilm formation ability under in vitro conditions, which significantly reduces the leaf lesion area and disease index of *Euonymus japonicus* [48].

In addition, plant growth-promoting rhizobacteria can also induce plant resistance, and plant defense enzyme activities are important indicators of plant resistance. The activities of plant polyphenol oxidase (PPO), peroxidase (POD), and superoxide dismutase (SOD) are closely related to increased plant disease resistance.

Polyphenol oxidase (PPO) is a copper-containing enzyme involved in catalyzing the oxidation of numerous phenolic compounds into antimicrobial quinones and the lignification of plant cells [49]. Peroxidase (POD) plays a crucial role as a secondary response enzyme in the elimination of reactive oxygen species (ROS) in plants. It can catalyze the generation of H_2_O and ·OH from H_2_O_2_ via catalase (CAT) and iron [50,51]. SOD catalyzes the production of H2O2 from reactive oxygen species, thereby increasing plant resistance. This study revealed that inoculation with *Bacillus velezensis* strain F9 increased the SOD enzyme activity of cucumber leaves in potting and field trials, thereby resisting the damage caused by cucumber Fusarium wilt. Similarly, *Bacillus velezensis* strain X5 can also increase the defense enzyme activities of tomato leaves to increase the resistance of tomato plants to bacterial wilt [52].

Notably, plant growth-promoting rhizobacteria can also affect the physicochemical properties of soil and the soil microbial community. Soil enzyme activity can indirectly reflect changes in soil microbial community abundance. Soil enzyme activity plays multiple roles, including nutrient cycling, energy conversion, and the promotion of increases in crop yield. Soil enzyme activity has been reported to be associated with plant growth, disease resistance, and increased nutrient contents. This study revealed that the *Bacillus velezensis* strain F9 increased the activity of neutral phosphatase in the soil during potting and field trials. However, further exploration of its impact on the soil microbial community is needed.

## Figures and Tables

**Figure 1 microorganisms-12-01882-f001:**
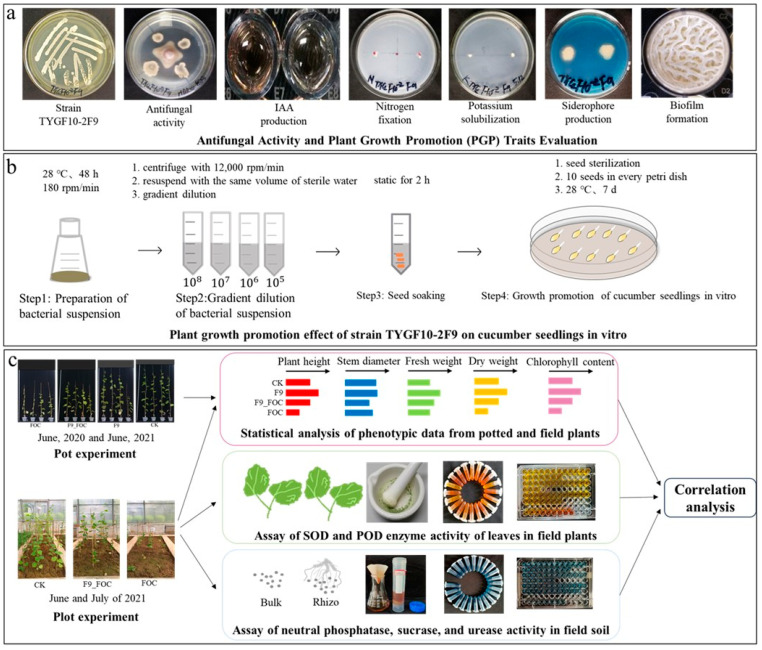
Flowchart for evaluating the effectiveness of *Bacillus velezensis* F9 in controlling cucumber wilt disease. (**a**) Antifungal activity and plant growth promotion (PGP) trait evaluation; (**b**) Plant growth promotion effect of strain F9 on cucumber seedlings in vitro; (**c**) Evaluating the efficacy and investigating the mechanism of strain F9 in controlling cucumber wilt in pot and field experiments.

**Figure 2 microorganisms-12-01882-f002:**
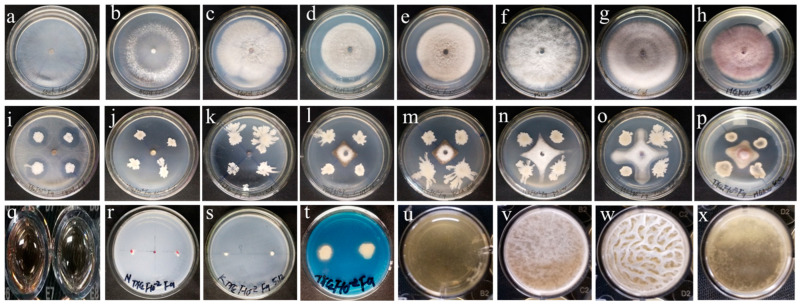
Evaluation of the antimicrobial and growth-promoting characteristics of strain F9. (**a**–**h**) represent pathogen control; (**i**–**p**) represent the inhibitory effect of strain F9 against *Rhizoctonia solani* (**i**); *Sclerotinia sclerotiorum* (**j**); *Botrytis cinerea* (**k**); *Gaeu-mannomyces graminsis (sacc.) Arx & Olivier Var tritici J. Walker* (**l**); *Alteraria alternate* (**m**); *Botryospuaeria berengeriana* (**n**); *Fusarium oxysporum f.* sp. *niveum* (**o**); *Fusarium oxysporum f.* sp. *cucumebrium Owen* (**p**). (**q**) IAA production assay; (**r**) nitrogen fixation assay; (**s**) potassium solubilization activity assay; (**t**) siderophore activity assay; (**u**–**x**) biofilm formation capability assay, where u is the blank control in LB medium; (**v**) biofilm in LBG medium; (**w**) biofilm in LBGM; and (**x**) biofilm in LB medium.

**Figure 3 microorganisms-12-01882-f003:**
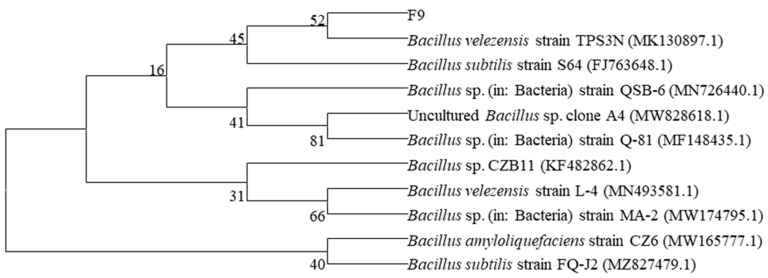
Phylogenetic tree of strain F9 on the basis of 16S rDNA by neighbor-joining method. Bootstrap values are based on 1000 repeats.

**Figure 4 microorganisms-12-01882-f004:**
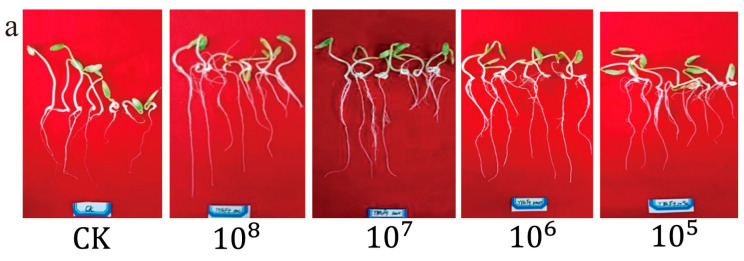
Growth-promoting effect of strain F9 on cucumber seedlings in vitro. (**a**) Growth promotion effect in vitro; (**b**) germination ratio; (**c**) seedling length; (**d**) root length; (**e**) fresh weight. Different letters represent significant differences at the 0.05 level according to ANOVA. The error bar represents the SE.

**Figure 5 microorganisms-12-01882-f005:**
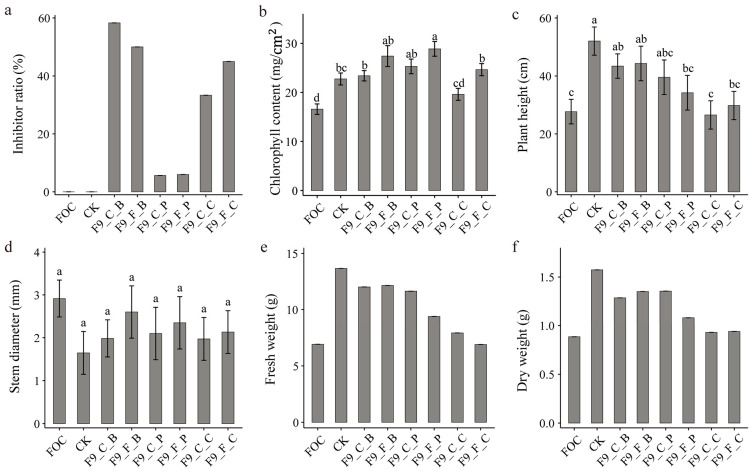
Disease control and growth promotion effects of strain F9 on cucumber strain F9 in June 2020. (**a**) inhibition ratio, (**b**) chlorophyll content, (**c**) plant height, (**d**) stem diameter, (**e**) fresh weight, and (**f**) dry weight. Each treatment was replicated 4 times, and the data are presented as the means ± SEs. Different lowercase letters indicate significant differences between treatments at the 0.05 level. During data analysis, all replicates were pooled for disease index, fresh weight, and dry weight calculations. Therefore, no replicate values were available for statistical significance testing.

**Figure 6 microorganisms-12-01882-f006:**
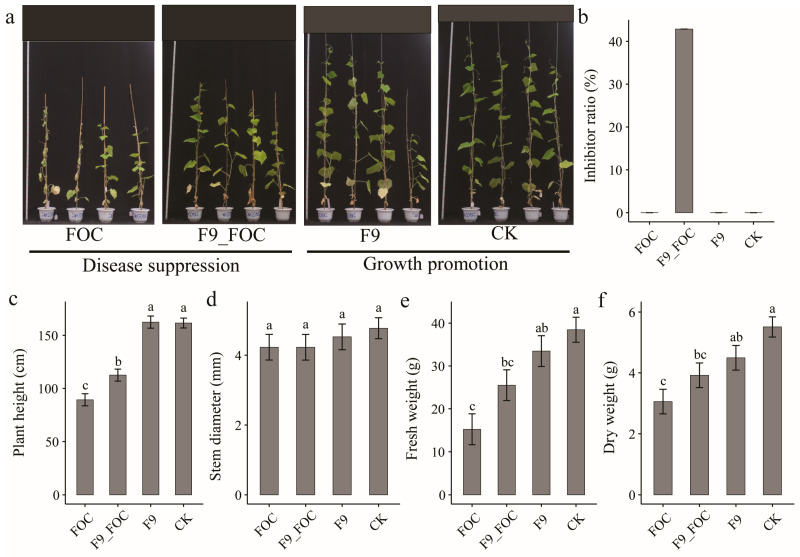
Disease control and growth promotion effects of strain F9 on cucumber plants in June 2021. (**a**) Pot pictures, (**b**) inhibition ratio, (**c**) plant height, (**d**) stem diameter, (**e**) fresh weight and (**f**) dry weight. Each treatment was replicated 4 times, and the data are presented as the means ± SEs. Different lowercase letters indicate significant differences between treatments at the 0.05 level. The inhibition ratio represents individual values obtained by combining and analyzing 4 replicates of cucumber plants; therefore, mean and standard error calculations cannot be performed.

**Figure 7 microorganisms-12-01882-f007:**
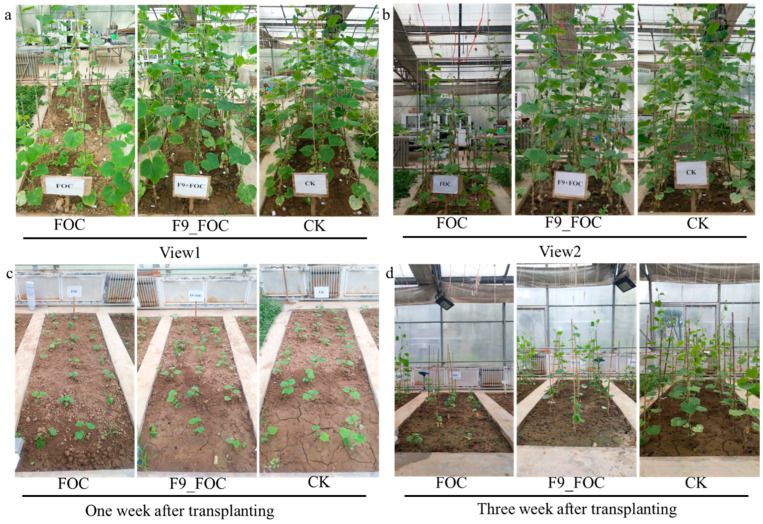
Inhibitory effect of strain F9 on Fusarium wilt in cucumber in the field. (**a**,**b**) are pictures of different views in June 2021; (**c**,**d**) are pictures of different times in July 2021.

**Figure 8 microorganisms-12-01882-f008:**
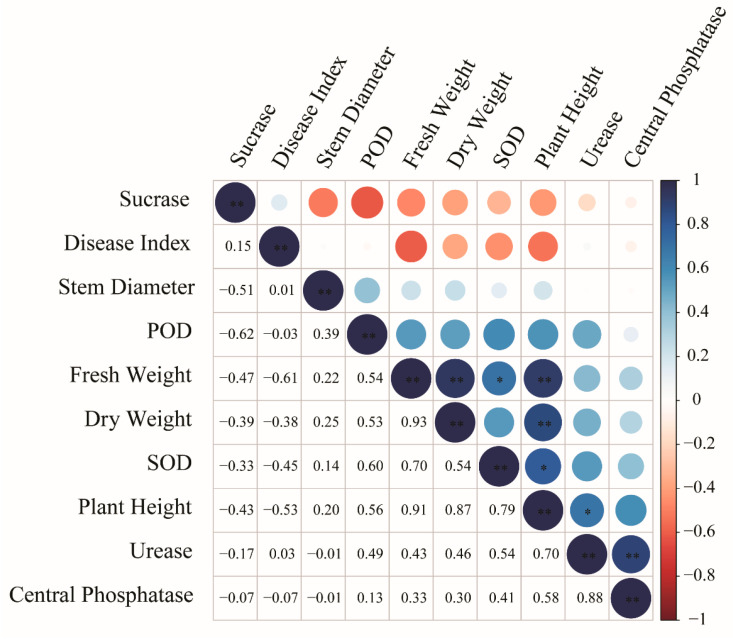
Correlation heatmap of cucumber plant phenotypic traits in the plot experiment. The top right corner represents the heatmap with significant markers, whereas the bottom left corner displays the correlation coefficient (r) values calculated by person method. “*” represents significance at the 0.05 level, and “**” represents significance at the 0.01 level.

**Table 1 microorganisms-12-01882-t001:** Inhibitory rate of *Bacillus velezensis* F9 on eight phytopathogens.

Strain	F9		
Name	Fungi Semi-Diameter (cm)	Inhibitory Rate (%)	Zone Width (cm)
*R. solani*	1.51 ± 0.04	66.39 ± 0.79	0.35 ± 0.05
*S. sclerotiorum*	0.49 ± 0.23	88.18 ± 5.53	1.33 ± 0.63
*B. cinerea*	0.53 ± 0.1	85.24 ± 2.87	1.08 ± 0.35
*G. graminsis*	0.69 ± 0.15	82.03 ± 4.00	0.77 ± 0.22
*A. alternate*	1.06 ± 0.07	66.80 ± 2.17	0.61 ± 0.29
*B. berengeriana*	1.3 ± 0.08	66.55 ± 2.04	0.66 ± 0.17
*F. oxysporum f.* sp. *niveum*	1.48 ± 0.12	62.66 ± 2.95	0.30 ± 0.10
*F. oxysporum f.* sp. *cucumerinum*	1.36 ± 0.15	64.07 ± 3.84	0.35 ± 0.20

**Table 2 microorganisms-12-01882-t002:** PGP traits of *Bacillus velezensis* F9.

Strain	IAA (µg/mL)	Siderophore (mm)	Nitrogen Fixation	Dissolved Organic Phosphorus	Decomposing Potassium
F9	5.97 ± 1.75	2	+	−	−

**Table 3 microorganisms-12-01882-t003:** Inhibition efficiency of strain F9 on Fusarium wilt of cucumber in field (June).

Treatment	IR (%)	Plant Height (cm)	Stem Diameter (mm)	Fresh Weight (g)	Dry Weight (g)
FOC	-	144.62 ± 66.36 b	6.01 ± 0.73 a	73.97 ± 60.13 b	9.97 ± 5.67 b
F9-FOC	44.95	257.56 ± 68.55 a	6.05 ± 0.9 a	146.45 ± 82.34 a	15.64 ± 6.46 a
CK	-	226.43 ± 74.49 a	5.91 ± 0.87 a	147.97 ± 76.49 a	13.8 ± 6.55 ab

Note: Data are mean ± SE. Different letters represent significant difference at 0.05 level.

**Table 4 microorganisms-12-01882-t004:** Inhibition efficiency of strain F9 on Fusarium wilt of cucumber in field (July).

Treatment	IR (%)	Plant Height (cm)	Stem Diameter (mm)	Fresh Weight (g)	Dry Weight (g)
FOC	-	38.29 ± 7.78 b	3.39 ± 0.24 b	3.38 ± 0.27 b	7.49 ± 3.07 b
F9-FOC	33.99	87.59 ± 63.02 a	4.41 ± 1.42 a	4.43 ± 1.32 a	28.41 ± 30.32 a
CK	-	120.39 ± 33.09 a	4.19 ± 0.81 ab	4.19 ± 0.81 a	42.7 ± 17.64 a

Note: Data are mean ± SE. Different letters represent significant difference at 0.05 level.

**Table 5 microorganisms-12-01882-t005:** Effect of strain F9 on soil enzyme activity of cucumber in field.

Treatment	Central Phosphatase (U/g)	Sucrase (U/g)	Urease (U/g)
CK	2.19 ± 0.75 a	18.23 ± 4.46 a	0.08 ± 0.01 a
FOC	2.11 ± 0.87 a	18.95 ± 6.29 a	0.08 ± 0.01 a
F9_FOC	2.58 ± 0.88 a	17.38 ± 4.78 a	0.09 ± 0.01 a

Note: Data are mean ± SE. Different letters represent significant difference at 0.05 level.

**Table 6 microorganisms-12-01882-t006:** Effect of strains F9 on defense enzyme activity of cucumber leaf.

Time	Treatment	SOD (U/g)	POD (U/g)
2021.06	CK	2.06 ± 0.30 a	16,366.00 ± 14,088.36 b
	F9_FOC	2.13 ± 0.28 a	15,127.64 ± 8515.44 b
	FOC	1.94 ± 0.12 a	26,575.29 ± 9369.72 a
2021.07	CK	1.98 ± 0.32 a	30,426.77 ± 21,226.04 a
	F9_FOC	2.08 ± 0.26 a	18,106.94 ± 8853.32 b
	FOC	1.96 ± 0.09 a	32,085.20 ± 16,453.86 a

Note: Significant differences in the table are observed between the three treatments within each field plot. Data are mean ± SE.

## Data Availability

The original contributions presented in the study are included in the article, further inquiries can be directed to the corresponding author.

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
