# Peer review of "Evaluation of Bacillus velezensis F9 for Cucumber Growth Promotion and Suppression of Fusarium wilt Disease"

_microorganisms, 2024, doi:10.3390/microorganisms12091882_

Round 1
Reviewer 1 Report
Comments and Suggestions for Authors
I thought the paper was well written and the experiments appropriate for the question. There were a few details missing from the methods and results that are important but I am sure these will be easy edits for the authors. I also have minor editorial suggestions.
All concerns are listed below in more or less the order they appear in the manuscript.
Line 100: change "alternate" to "alternata"
Line 116: Was the measurement to the greatest distance spanned by the mycelia, the shortest distance, or what was deemed to be the average distance?
Line 118: If this measurement is truly a "rate" then there should be a time elapsed comparison somewhere in the formula. From reading the manuscript, I do not believe the authors measured a rate but rather a difference. Perhaps the "inhibition rate" should be changed to "% inhibition" throughout the manuscript.
Line 198: Provide a citation explaining your disease index or describe how the disease index was measured. This response variable measurement could not be replicated as currently described.
Figure 2. Indicate in the figure caption which picture pairs belong to which pathogen in parentheses following the pathogen names.
Figure 3. This looks like a UPGMA tree. Please indicate the type of tree (e.g. neighbor joining, UPGMA, maximum likelihood, etc.) in the figure caption.
Figures 4,5,6, and Tables 3,4,5,6. The authors report statistical results in the form of post-hoc pairwise comparisons, but they do not report the results of ANOVAs. Please include the ANOVA test results, either in the figure legend, the figure itself, or a supplementary file that contains the statistical results.
Tables 3&4. In the table titles change "efficient" to "efficiency"
Section 3.5. I assume that the authors used Pearson correlations and not Spearman Rank Correlations. Please indicate which test was used.
Line 447: Italicize "Pseudomonas"
Line 497: Italicize "Paenibacillus polymyxa"
Line 548: Italicize Eunonymus japonicus
Author Response
Comments 1: I thought the paper was well written and the experiments appropriate for the question. There were a few details missing from the methods and results that are important but I am sure these will be easy edits for the authors. I also have minor editorial suggestions.
All concerns are listed below in more or less the order they appear in the manuscript.
Response1: Thank you very much for your positive comments.
Comments 2: Line 100: change "alternate" to "alternata"
Response2: We are so sorry for this error and have corrected it in the article.
Comments 3: Line 116: Was the measurement to the greatest distance spanned by the mycelia, the shortest distance, or what was deemed to be the average distance?
Response3: It is the average distance four values obtained through the cross-crossing method.
Comments 4: Line 118: If this measurement is truly a "rate" then there should be a time elapsed comparison somewhere in the formula. From reading the manuscript, I do not believe the authors measured a rate but rather a difference. Perhaps the "inhibition rate" should be changed to "% inhibition" throughout the manuscript.
Response4: Thank you for this very insightful comment. We have changed all instances of "inhibition rate" into "inhibition ratio" in both the text and figures of the article.
Comments 5: Line 198: Provide a citation explaining your disease index or describe how the disease index was measured. This response variable measurement could not be replicated as currently described.
Response5: We are very sorry for this error and have corrected it in the article. Disease index (%) = (∑(disease severity level × number of diseased plants)) / (highest disease severity level × total number of plants) × 100. Control efficacy was determined by the formula (Control DI - Treatment DI)/Control DI× 100%.
Comments 6: Figure 2. Indicate in the figure caption which picture pairs belong to which pathogen in parentheses following the pathogen names.
Response6: Thank you for pointing out the deficiency in Figure 2's caption. We have now labeled each image with the name of the pathogenic fungus it represents.
Comments 7: Figure 3. This looks like a UPGMA tree. Please indicate the type of tree (e.g. neighbor joining, UPGMA, maximum likelihood, etc.) in the figure caption.
Response 7: Thank you for this very insightful comment. We constructed a phylogenetic tree using the neighbor-joining method, and the caption has been updated to include "neighbor-joining" for clarity.
Comments 8: Figures 4,5,6, and Tables 3,4,5,6. The authors report statistical results in the form of post-hoc pairwise comparisons, but they do not report the results of ANOVAs. Please include the ANOVA test results, either in the figure legend, the figure itself, or a supplementary file that contains the statistical results.
Response 8: Thanks for your suggestion. However, we do analyze using ANOVA and the mean values were compared using LSD multiple range test with a significance level set at p<0.05 in R 4.3.1 software. Therefore, all of the results are the ANOVA test results. Please feel free to correct me if I have misunderstood your feedback.
Comments 9: Tables 3&4. In the table titles change "efficient" to "efficiency"
Response 9: Thank you for pointing out the deficiency in titles of Tables 3&4. We have changed change "efficient" to "efficiency" in titles of Tables 3&4.
Comments 10: Section 3.5. I assume that the authors used Pearson correlations and not Spearman Rank Correlations. Please indicate which test was used.
Response 10: We gratefully appreciate for your constructive suggestion. We evaluated the correlation of cucumber plant phenotypic traits using Pearson correlations. We have added supplementary information in the "2.10. Statistical Analyses" section and the caption of Figure 8.
Comments 11: Line 447: Italicize "Pseudomonas"
Response 11: We are very sorry for this error and have corrected it in the article.
Comments 12: Line 497: Italicize "Paenibacillus polymyxa"
Response 12: We are very sorry for this error and have corrected it in the article.
Comments 13: Line 548: Italicize Eunonymus japonicus
Response 13: We are very sorry for this error and have corrected it in the article.
Reviewer 2 Report
Comments and Suggestions for Authors
Evaluation of Bacillus velezensis TYGF10-2F9 for cucumber
growth promotion and suppression of Fusarium wilt disease
General remarks:
The paper is very extensive and contains a lot of information. I would like to praise the authors for the concept and the detailed paper, as well as the topic, which is always current. Also, I would like to praise the authors for the comprehensive graphic abstract.
A general objection to the authors is that in the text of the manuscript they mark the Bacillus strain in two ways. Namely, for the microorganism Bacillus velezensis TYGF10-2F9, they use TYGF10-2F9 in some places, and F9 in another place. It is especially confusing when they use F9 in the table and then explain it in the text as TYGF10-2F9. In order not to be confusing, I suggest to make the text uniform and to use only one of these two markings.
Other suggestions for changing or supplementing the text will be written below by items:
Specific remarks:
Page 3, Lines 99-107 - Write under what conditions and on what medium the mentioned microorganisms are stored in the Culture Collections.
Page 3, Line 112 – Mycelium of which microorganisms? Explain to make it clear to the reader.
Page 3, Lines 113-114 - Are they 4 different test isolates or is it one in 4 replicates. Explain in more detail.
Page 4, Line 124 – LB media; Media (plural) or medium (singular)? Explain what medium it is. What is the composition or state the manufacturer if it is a commercial medium.
Page 4, Line 142 – Alexandrov media; Media or medium? Explain what medium it is. What is the composition or state the manufacturer if it is a commercial medium.
Page 4, Line 146 - ACCC55 nitrogen-free media; Media or medium? Explain what medium it is. What is the composition or state the manufacturer if it is a commercial medium.
Page 4, Line 152 - Explain each abbreviation that appears for the first time in the text. Also, explain the difference between the mentioned media.
Page 4, Line 166 – PBS buffer; Explain the abbreviation.
Page 5, Line 198 - How did you determine the disease index?
Page 5, Lines 211-227 - Has anyone else used this methodology? Add references.
Also, how was the enzyme activity calculated?
Page 6, Lines 231-242 - How did you calculate the enzyme activity?
Page 6, Lines 255-256 - Modify the labeling as suggested for Figure 2.
Page 7, Figure 2 - It is clear to me that the authors wanted to present the top row as controls (a-0) and the bottom row (b-p) as the activity of F9 on pathogens, however, this way of labeling confused me a bit if we consider the alphabet. I would ask the authors to change the labeling so that the upper row is, for example, from a-h and the lower logically from i-p. The third row is logically continued, but the labeling of the first 2 rows is confusing.
Page 8, Figure 4 – b fermentation rate or germination rate?
Page 8, Lines 316-117 - Nowhere is it indicated which test was used to determine the level of statistical significance.
Page 9, Lines 320-324 - In my opinion, this explanation should be written in the Material and Methods section. Also, it is confusing that you used the letter C 2 times in the marking.
Page 9, Lines 330-332 - Did you determine this visually or statistically, since I don't see significance levels in Figure 5b?
Page 9, Lines 334-335 - If I am not mistaken, the measurement of chlorophyll content is not mentioned in the Material and Methods section!?
Page 9, Line 341 - Figure 5 a, d, e, f, g; Delete 5a.
Page 9, Lines 347-350 - I see no reason why the mean values ​​cannot be determined.
Page 14, Line 447 – Pseudomonas italic.
General opinion:
I suggest that the paper be accepted for publication after a proposed revision.
Best regards,
Reviewer
Author Response
Comments 1: General remarks: The paper is very extensive and contains a lot of information. I would like to praise the authors for the concept and the detailed paper, as well as the topic, which is always current. Also, I would like to praise the authors for the comprehensive graphic abstract.
Response 1: Thank you very much for your positive comments and professional review work on our article.
Comments 2: A general objection to the authors is that in the text of the manuscript they mark the Bacillus strain in two ways. Namely, for the microorganism Bacillus velezensis TYGF10-2F9, they use TYGF10-2F9 in some places, and F9 in another place. It is especially confusing when they use F9 in the table and then explain it in the text as TYGF10-2F9. In order not to be confusing, I suggest to make the text uniform and to use only one of these two markings.
Other suggestions for changing or supplementing the text will be written below by items:
Response 2: Thank you for this very insightful comment. We have changed all instances of " TYGF10-2F9" into "F9" in both the text and figures of the article.
Specific remarks:
Comments 3: Page 3, Lines 99-107 - Write under what conditions and on what medium the mentioned microorganisms are stored in the Culture Collections.
Response 3: Thank you for pointing out the information deficiency of strain storage. We have add the information to “2.1. Fungal pathogens, biocontrol strains and plants”, as follows: The fungal pathogens were cryopreserved in potato dextrose broth (PDB) medium containing 20% glycerol at -80℃. The strain F9 was cryopreserved in LB medium containing 20% glycerol at -80℃.
Comments 4: Page 3, Line 112 – Mycelium of which microorganisms? Explain to make it clear to the reader.
Response 4: We gratefully appreciate for your constructive suggestion. We have provided a detailed description in the paper, where "mycelia" refers to the eight plant pathogenic fungi mentioned in section 2.1.
Comments 5: Page 3, Lines 113-114 - Are they 4 different test isolates or is it one in 4 replicates. Explain in more detail.
Response 5: I apologize for the confusion caused by the unclear description. This study only includes one biocontrol strain, Bacillus velezensis F9. The reference should be understood as four repetitions of the experiment using the same strain F9. We have provided a detailed description of this in the paper.
Comments 6: Page 4, Line 124 – LB media; Media (plural) or medium (singular)? Explain what medium it is. What is the composition or state the manufacturer if it is a commercial medium.
Response 6: It should be LB medium. It is used for bacterial culture. LB medium is composed of 10 g of tryptone, 5 g of yeast extract, and 5 g of NaCl per liter. We have provided a detailed description of this in the paper.
Comments 7: Page 4, Line 142 – Alexandrov media; Media or medium? Explain what medium it is. What is the composition or state the manufacturer if it is a commercial medium.
Response 7: It should be Alexandrov medium. Alexandrov medium is composed of 5 g Glucose, 0.5 g MgSO4·7H2O, 0.0005 g FeCl3, 0.1 g CaCl2, 0.2 g Al2K2Na2O15Si5, 0.2 g Ca3(PO4)2, 1000 mL distilled water, pH=7.0-7.2. We have provided a detailed description of this in the paper.
Comments 8: Page 4, Line 146 - ACCC55 nitrogen-free media; Media or medium? Explain what medium it is. What is the composition or state the manufacturer if it is a commercial medium.
Response 8: It should be ACCC55 nitrogen-free medium. ACCC55 nitrogen-free medium is composed of 10 g Sucrose, 0.12 g NaCl, 0.5 g K2HPO4·3H2O, 1 g CaCO3, 0.2 g MgSO4·7H2O, 20 g agar, 1000 mL distilled water, pH=7.0-7.2. We have provided a detailed description of this in the paper.
Comments 9: Page 4, Line 152 - Explain each abbreviation that appears for the first time in the text. Also, explain the difference between the mentioned media.
Response 9: Each abbreviation is the name of different media. LB medium, LBG medium (LB plus 1% [vol/vol] glycerol), LBM medium (LB plus 0.1 mM MnSO4), or LBGM medium (LB plus 1% [vol/vol] glycerol and 0.1 mM MnSO4). We have provided a detailed description of this in the paper.
Comments 10: Page 4, Line 166 – PBS buffer; Explain the abbreviation.
Response 10: PBS is the shorter of phosphate buffer saline. We have provided a detailed description of this in the paper.
Comments 11: Page 5, Line 198 - How did you determine the disease index?
Response 11: We have added references for the statistics and calculation of disease index and control efficacy in the article. In brief, disease severity was assessed using a 0-4 disease scale; 0 = leaf asymptomatic; 1 = leaf wilting below 1/4 of cucumber seedling; 2 = leaf wilting in 1/4 to 1/2 of cucumber seedling; 3 = leaf wilting above 1/2 of cucumber seedling; 4 = the whole plant was wilted and died. The disease index was calculated using DI = [[(0×N0) + (1×N1) + (2×N2) + (3×N3) + (4×N4)]/T×4]×100, where N is the number of cucumber seedlings for each disease score and T is the total number of cucumber seedlings. Control efficacy = (DI of control - DI of treatment)/DI of control×100%. Additionally, the reference includes methods for measuring cucumber chlorophyll content, plant height, stem diameter, fresh weight, and dry weight.
Comments 12: Page 5, Lines 211-227 - Has anyone else used this methodology? Add references. Also, how was the enzyme activity calculated?
Response 12: We have added references for the methodology and calculation of SOD and POD in the article.
The formula of SOD is as follows:
where: Ao - Absorbance of the control tube at 560 nm under light conditions; As - Absorbance of the sample tube at 560 nm; Vt - Total volume of sample extract (mL); Vs - Volume of crude enzyme solution used for measurement (mL); FW - Fresh weight of the sample (g); t - Duration of color reaction under light (min).
The formula of POD is as follows:
where: ΔA470 - Change in absorbance over reaction time; FW - Fresh weight of leaves (g); t - Reaction time (min); Vt - Total volume of enzyme extract (mL); VS - Volume of enzyme solution used for measurement (mL).
Comments 13: Page 6, Lines 231-242 - How did you calculate the enzyme activity?
Response 13: We gratefully appreciate for your constructive suggestion. While we have provided the definition of enzyme activity measurement in the text and shown the derivation of the formula, your suggestion of presenting the formula directly would be more intuitive and concise. The calculation formulas of phosphatase activity, urease activity and sucrase activity as follows:
- Phosphatase activity = (A_sample - A_soil-free - A_matrix-free) × V × n / m
Where: A_sample: Phenol concentration (mg) in the sample, determined from the standard curve using the sample absorbance value; A_soil-free: Phenol concentration (mg) in the soil-free control, determined from the standard curve using the soil-free control absorbance value; A_matrix-free: Phenol concentration (mg) in the matrix-free control, determined from the standard curve using the matrix-free control absorbance value; V: Volume of color reagent (mL); n: Dilution factor, calculated as the volume of extractant solution / volume of filtered solution; m: Dry weight of soil (g).
- Urease activity = (A_sample - A_soil-free - A_matrix-free) × V × n / m
Where: A_sample: NH3-N concentration (mg) in the sample, determined from the standard curve using the sample absorbance value; A_soil-free: NH3-N concentration (mg) in the soil-free control, determined from the standard curve using the soil-free control absorbance value; A_matrix-free: NH3-N concentration (mg) in the matrix-free control, determined from the standard curve using the matrix-free control absorbance value; V: Volume of color reagent (mL); n: Dilution factor, calculated as the volume of extractant solution / volume of filtered solution; m: Dry weight of soil (g).
- Sucrase activity = (A_sample - A_soil-free - A_matrix-free) × n / m
Where: A_sample: Glucose concentration (mg) in the sample, determined from the standard curve; A_soil-free: Glucose concentration (mg) in the soil-free control, determined from the standard curve; A_matrix-free: Glucose concentration (mg) in the matrix-free control, determined from the standard curve; n: Dilution factor; m: Dry weight of soil (g).
Comments 14: Page 6, Lines 255-256 - Modify the labeling as suggested for Figure 2.
Response 14: We gratefully appreciate for your constructive suggestion. We have modified the label according to your suggestion.
Comments 15: Page 7, Figure 2 - It is clear to me that the authors wanted to present the top row as controls (a-0) and the bottom row (b-p) as the activity of F9 on pathogens, however, this way of labeling confused me a bit if we consider the alphabet. I would ask the authors to change the labeling so that the upper row is, for example, from a-h and the lower logically from i-p. The third row is logically continued, but the labeling of the first 2 rows is confusing.
Response 15: We gratefully appreciate for your constructive suggestion. We have modified the label according to your suggestion.
Comments 16: Page 8, Figure 4 – b fermentation rate or germination rate?
Response 16: It should be germination ratio. Thank you for pointing out the deficiency.
Comments 17: Page 8, Lines 316-117 - Nowhere is it indicated which test was used to determine the level of statistical significance.
Response 17: We gratefully appreciate for your constructive suggestion. We have added the test methods in “2.10. Statistical Analyses”.
Comments 18: Page 9, Lines 320-324 - In my opinion, this explanation should be written in the Material and Methods section. Also, it is confusing that you used the letter C 2 times in the marking.
Response 18: We concur with your insightful observation and have incorporated it into Section 2.7 of the manuscript. To avoid ambiguity, we have also revised the numbering system, changing the bacterial suspension designation from “C” to “F9_C” to distinguish it from the treatment mode designation “C”.
Comments 19: Page 9, Lines 330-332 - Did you determine this visually or statistically, since I don't see significance levels in Figure 5b?
Response 19: We gratefully appreciate for your constructive suggestion. We have opted for a more precise descriptor by replacing "significant" with "obvious" in the sentence.
Comments 20: Page 9, Lines 334-335 - If I am not mistaken, the measurement of chlorophyll content is not mentioned in the Material and Methods section!?
Response 20: We gratefully appreciate for your constructive suggestion. We have added references for the measuring and calculation of cucumber chlorophyll content, plant height, stem diameter, fresh weight, dry weight, disease index and control efficacy.
Comments 21: Page 9, Line 341 - Figure 5 a, d, e, f, g; Delete 5a.
Response 21: We concur with your insightful observation and have deleted it.
Comments 23: Page 9, Lines 347-350 - I see no reason why the mean values ​​cannot be determined.
Response 23: We gratefully appreciate for your constructive suggestion. During data analysis, actually, all replicates were pooled for disease index, fresh weight, and dry weight calculations. Therefore, no replicate values were available for statistical significance testing.
Comments 24: Page 14, Line 447 – Pseudomonas italic.
Response 24: We are very sorry for this error and have corrected it in the article.
Comments 25: General opinion: I suggest that the paper be accepted for publication after a proposed revision.
Response 25: Thank you very much for your positive comments and professional review work on our article.
Round 2
Reviewer 2 Report
Comments and Suggestions for Authors
Evaluation of Bacillus velezensis F9 for cucumber growth promotion and suppression of Fusarium wilt disease
I am very satisfied with the answers and changes made by the authors in the article.
I would like the authors to include in the paper the disease scale (just short explanation) and the method of calculating the enzyme activity.
Best regards,
Rewiever
Author Response
Comments 1: I am very satisfied with the answers and changes made by the authors in the article.
Response 1: Thank you very much for your positive comments and professional review work on our article.
Comments 2: I would like the authors to include in the paper the disease scale (just short explanation) and the method of calculating the enzyme activity.
Response 2: We gratefully appreciate for your constructive suggestion. We have added the explanation of the disease scale and the method of calculating the disease index, the control efficacy and the enzyme activity in the article.
